# Autogenous Oxidation/Reduction of Polyaniline in Aqueous Sulfuric Acid

**Amrita Singh [1], Ravindra Kale [1], Arindam Sarkar [2], Vinay Juvekar [2] and Asfiya Contractor [1],***

[1] Department of Fibers and Textile Processing Technology, Institute of Chemical Technology, Mumbai 400019, India; amrita.sentient@gmail.com (A.S.); rd.kale@ictmumbai.edu.in (R.K.)
[2] Department of Chemical Engineering, Indian Institute of Technology Bombay, Mumbai 400076, India; asarkar@che.iitb.ac.in (A.S.); vaj@iitb.ac.in (V.J.)
* Correspondence: asfiyacontractor@gmail.com

**Abstract:** In this work, we have shown through open circuit potential experiments that in aqueous sulfuric acid solutions, a thick polyaniline film undergoes autogenous oxidation when reduced below a threshold potential and autogenous reduction when oxidized above the threshold potential. This phenomenon is associated with the high resonance stability of polarons in long polyaniline chains present in thicker films. We have determined the rates of these reactions using a linear sweep chronopotentiometry technique. We propose that the oxidation reaction of polyaniline produces polarons with a concomitant reduction of hydrogen ions to hydrogen radicals, which further combine with each other to produce the hydrogen molecule in the absence of dissolved oxygen. On the other hand, at high potentials polarons are reduced with the concomitant oxidation of water to hydroxyl radicals. Both the radicals are stabilized due to the interaction of their unpaired electrons with pi-electrons of the aromatic rings of the polymer backbone. At the equilibrium value of the open circuit potential, both the hydrogen radicals and hydroxyl radicals are generated at equal rates and react with each other to form water.

**Keywords:** intrinsically conducting polymers; polyaniline; highly conjugated polymer; autogenous oxidation/reduction; open circuit potential; equilibrium potential





## 1. Introduction

Intrinsically conducting polymers possess dopable sites which can be doped/dedoped either by using oxidizing/reducing agents or by the application of the external electric potential. The oxidized sites of the polymer exist as radical cations (polarons and bipolarons). The stability of polarons depends on the extent of conjugation because conjugation causes the delocalization of charges on the polarons. The extent of conjugation depends on the number of energy states available for the charges associated with polarons. With the increase in the chain length of the polymer, the number of energy states increases [1]. When conjugation is sufficiently high, the polymer gains the ability to perform autogenous reactions in which the polymer can reduce (oxidize) a reactive species present in the solution at the expense of its own oxidation (reduction). This would cause an increase (decrease) in the potential of the polymer when it is kept under the open circuit condition in a solution containing the reactant. In the present work, we observed that when a polyaniline film, electrochemically grown on the platinum electrode, is electrochemically reduced to a low potential ($-0.2\ V_{SCE}$) and then immersed in an aqueous solution of sulfuric acid, its open circuit potential increases with time until it attains a constant value (equilibrium potential). This implies that the polymer undergoes autogenous oxidation during this experiment (we call this a rising OCP experiment). On the other hand, when the potential of the electrode is electrochemically raised beyond the equilibrium potential and the electrode is allowed to relax under the open circuit condition, its potential decreases (we call this a falling OCP experiment) until it eventually attains the same equilibrium value as is observed during

the rising OCP experiment. Both oxidation and reduction reactions happen even under the inert atmosphere indicating there is no participation of dissolved oxygen in these reactions. The focus of the present work is to attempt to understand this behavior and also to elucidate certain aspects of its mechanism.

Conducting polyaniline films are used as electrodes in supercapacitors, renewable batteries and fuel cells [2–6]. These applications usually make use of thin polyaniline films [7–9]. Charging of these films under ambient conditions usually takes place by the oxygen reduction reaction (ORR) [10–13]. In this work, we report charging or polaron formation in polyaniline films in the absence of oxygen. We have shown that this reaction takes place in thick polyaniline films under a nitrogen blanket, that is, autogenous oxidation. Such thick polyaniline films which are self-doping can be used as mediators for oxidation and reduction reactions. Dilute acid can be used to charge the film (with the concomitant reduction of protons to hydrogen gas) and water can be used to reduce the film (with the concomitant oxidation of water to oxygen gas). Therefore, the film can not only be employed as a reducing agent but also as an oxidizing agent if charged electrochemically to open circuit potentials above its equilibrium potential.

## 2. Materials and Methods

### 2.1. Materials

Analytical grade aniline, sulfuric acid 98% (*w/w*), nitric acid 65% (*w/w*) and potassium iodide were purchased from Merck. Aniline was distilled to a colorless liquid before use. For the purpose of the standardization of polyaniline, the polymer powder (Mw = 15000 Da) purchased from Sigma Aldrich was used. The nitrogen used in these experiments had purity greater than 99.9 per cent.

### 2.2. Setup

The setup comprised of a single compartment three-electrode cell. The working electrode was a rectangular platinum plate of dimensions 18.9 mm $\times$ 8.8 mm $\times$ 0.5 mm (total geometric surface area 360 mm$^2$). A saturated calomel electrode (SCE) was used as the reference electrode and a cylindrical platinum gauze was the counter electrode. All potentials reported here are with respect to SCE. An electrochemical workstation (CH Instruments Model 600D) was used to control the electrical inputs and outputs. UV-VIS 160 (Shimadzu, Japan) was used for spectrophotometric analysis. Agitation of the solution was performed by the rotating cylinder (13 mm diameter) of an RDE apparatus at the speed of 2000 RPM. Since the surface of the liquid was exposed to air, the system can be considered as surface aerated. Many experiments were also conducted by bubbling oxygen free nitrogen into the solution.

### 2.3. Polymer Synthesis

The procedure for the electrodeposition was the same as that used previously by our group [14]. The platinum plate electrode was pre-treated and activated before use. The electrodeposition was performed using cyclic voltammetry in 100 mL of 0.5 M sulfuric acid containing 0.1 M aniline, at the scan rate of 50 mV s$^{-1}$. At the end of the deposition, the plate was washed thoroughly with Milli-Q water and subjected to further experiments. All subsequent experiments on the polyaniline film were conducted under the stirred condition using the cylinder of RDE, rotating at 2000 RPM. The surface mass density of the polymer was varied between 0.6 g m$^{-2}$ and 39 g m$^{-2}$ (based on the geometric surface area of the plate electrode).

### 2.4. Estimation of Mass of Deposited Polymer

The polymer was dissolved in 5 mL of concentrated nitric acid. The solution was diluted and the absorbance of the solution was measured at the wavelength of 345 nm (this peak was verified to be an oxidation product of aniline). A calibration chart was prepared by measuring the absorbance of solutions of different concentrations of polyaniline powder

(Sigma Aldrich, Bangalore, India) in nitric acid. To validate the calibration for the electrodeposited polymer, the polymer film was scraped from the plate electrode and weighed. It was then dissolved in nitric acid and the intensity of the UV peak measured after dilution. The mass of the polymer, estimated using the previously prepared calibration chart, tallied with the actual mass within 3% error.

## 3. Results

### 3.1. Characteristics of High Conjugation Polyaniline Films

Since the ability of polyaniline to undergo autogenous oxidation/reduction is substantial at high surface mass density of the polymer, we have synthesized the polyaniline films with high surface mass density. These films possess high degrees of conjugation and their current potential characteristics differ significantly compared to thin films.

Figure 1a,b shows cyclic voltammograms and cyclic coulograms of five polyaniline films with increasing mass density. It is clear from these figures that as we increase the mass density of polyaniline, the features of the voltammograms smoothen. The three familiar peaks of polyaniline, which are clearly visible in the lowest mass density film, are subdued at the intermediate mass density and completely vanish at the highest mass density. In the last case, the voltammogram attains a leaf-like shape (Figure 1a). This shape is indicative of merging of the energy bands and reduction in the spacing between consecutive energy levels allowing the almost continuous transition from one energy level to the next. This implies that there is a continuum of energy levels available for storage of charge and the polymer has the ability to oxidize/reduce ions from the medium at the expense of its own oxidation/reduction.

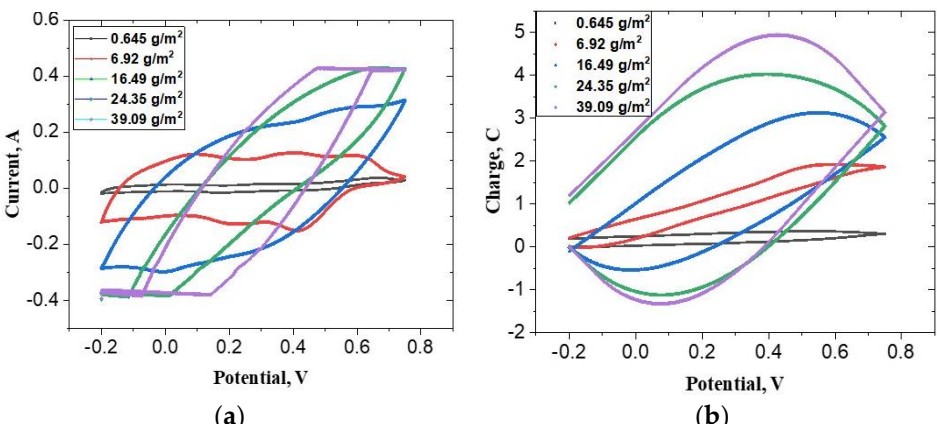

**Figure 1.** Cyclic voltammograms: (**a**) and cyclic coulogram plots; (**b**) of polyaniline films of different surface mass densities. Concentration of $H_2SO_4$ = 0.5 M, concentration of aniline = 0.1 M and temperature = 25 °C. Scan rate = 50 mV s$^{-1}$. All potentials are reported with respect to SCE.

The specific differential capacitance of the film, $c_f$, can be estimated from the coulograms using the equation,

$$c_f = (d\sigma/dE) \tag{1}$$

where, $\sigma$ is the surface charge density, and $E$, the electrode potential. It is seen from the coulograms that within the potential range relevant to autogenous oxidation ($-0.2$ to $0.4$ $V_{SCE}$), the charge density in thick films is a linear function of the potential. This indicates that the differential capacitance of the film is constant in this range.

### 3.2. Rising Open Circuit Potential Experiment

In this experiment, a polyaniline film was electrochemically reduced in 0.5 M sulfuric acid to $-0.2$ $V_{SCE}$ and its potential was subsequently monitored under the open circuit condition. A typical plot of the open circuit potential against time is shown in Figure 2.

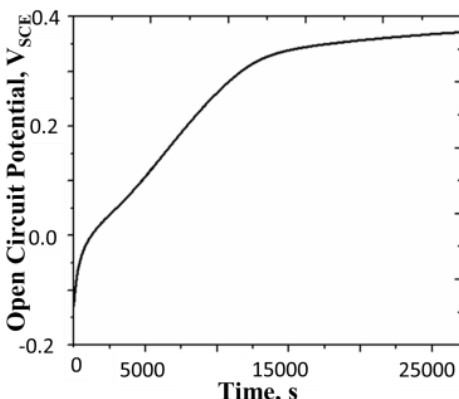

**Figure 2.** Rising open circuit potential of polyaniline film with initial potential of $-0.2$ V. Concentration of $H_2SO_4$ = 0.5 M, surface mass density of the film = 24.4 g m$^{-2}$, temperature = 25 °C. All potentials are reported with respect to SCE.

It is seen from the figure that for a very short initial period, the potential rises rapidly (this capacitive rise occurs due to formation of the electric double layer). This is followed by a regime where the potential varies linearly with time. During the last phase of evolution, the potential varies very slowly until it attains a constant value, which we call the equilibrium potential, $E_e$. The value of $E_e$ (measured after 18 h) is 0.39 $V_{SCE}$ for this film.

An increase in the potential of the electrode under the open circuit condition is a sign of the autogenous oxidation of polyaniline. The oxidation produces cation radicals (polarons) denoted by $-N^{·+}H-$. These impart a charge to the electrode, thereby raising its potential. We will discuss the probable mechanism for this reaction later.

The plot of the equilibrium potential versus polymer mass density is shown in Figure 3. It is seen that $E_e$ decreases with the increase in the mass density of the polymer, until it reaches a plateau of 0.39 V beyond the mass density of about 13 g m$^{-2}$. Assuming the bulk density of the film to be about 1000 kg m$^{-3}$, the thickness of the film in these experiments ranged from 0.6μm to 39 μm. Considering the length of one mer of polyaniline to be around 0.5 nm, this range film thickness corresponds to the variation in the chain length from 300 to 18,000 repeat units. From Figure 3, the point beyond which the equilibrium potential is independent of the chain length is estimated to be about 6500.

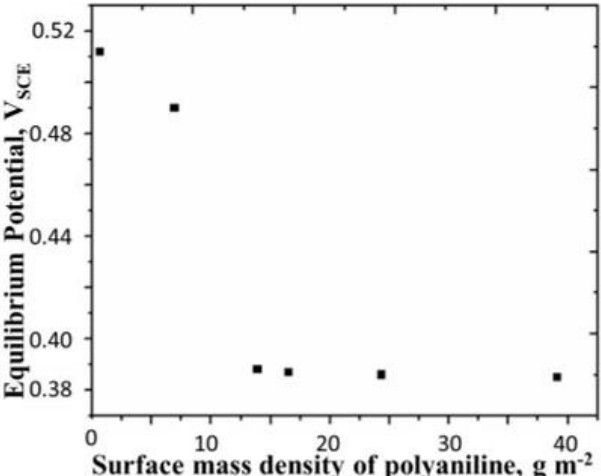

**Figure 3.** Equilibrium potential of polyaniline film versus surface mass density of polyaniline. Concentration of $H_2SO_4$ = 0.5 M, temperature = 25 °C. All potentials are reported with respect to SCE.

The film described in Figure 2 is reduced from its equilibrium potential down to $-0.2$ V using chronopotentiometry at a constant current of $-0.9$ mA. The resulting chronopotentiogram is shown in Figure 4a.

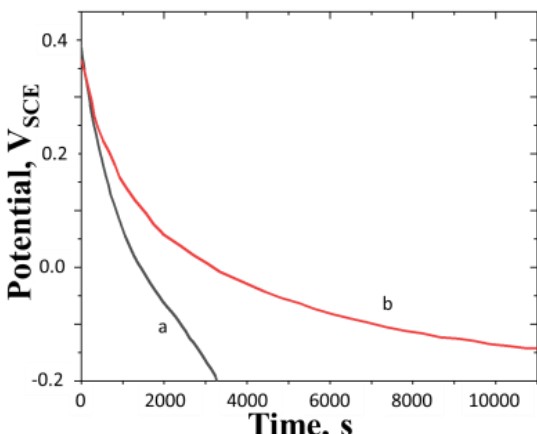

**Figure 4.** Chronopotentiograms for reduction of polyaniline films. (**a**) Black line: from equilibrium potential to $-0.2$ V at current density of $-2.25$ A m$^{-2}$; surface mass density of the film = 24.4 g m$^{-2}$; (**b**) Red line: constant current of $-8.75 \times 10^{-4}$ A m$^{-2}$; surface mass density of the film = 13.91 g m$^{-2}$. Concentration of H$_2$SO$_4$ = 0.5 M, temperature = 25 °C. All potentials are reported with respect to SCE.

Figure 4a has a point of inflection at around 1500 s, where the shape of the chronopotentiogram changes from concave to convex. The change in the curvature of the potentiogram is caused by the autogenous oxidation of the polymer. The resulting anodic current opposes the imposed cathodic current. At the point of inflexion, the autogenous anodic current has the highest magnitude.

From the total time of 3287 s, required for the cathodic reduction of the film from 0.38 V to $-0.2$ V at the current density of $-2.25$ A m$^{-2}$, the total charge contained in film is estimated to be 7396 C m$^{-2}$. This gives the integral capacitance of the film as 12.75 kFm$^{-2}$. Noting that the total mass of the film is 24.4 g m$^{-2}$, the apparent integral capacitance per unit mass is estimated to be 520 F g$^{-1}$. Based on the mass density of the film, the total amount of mers of aniline in the film is 0.268 mol m$^{-2}$. Hence the charge of 7396 C m$^{-2}$ corresponds to the oxidation of the film to the extent of 0.0766 mol m$^{-2}$, i.e., 28.6%. This indicates that the polyaniline film undergoes autogenous oxidation to a significant extent.

Figure 4b shows the chronopotentiogram of a thinner polyaniline film than in Figure 4a. Here, the cathodic reduction current density applied is four orders of magnitude lower. Yet, no inflection is observed in this curve indicating that the autogenous oxidation current is negligibly small. This clearly indicates that the thickness of the film has an important role to play in autogenous oxidation.

Since during our experiments, the solution was exposed to air and was also stirred, we expect the solution to be saturated with air. Hence, we suspect the oxygen reduction reaction (ORR) to be responsible for the rise of OCP. There are two possible reactions which can cause the oxidation of polyaniline in the potential range between $-0.2$ V$_{SCE}$ and 0.39 V$_{SCE}$. Both are associated with ORR.

$$O_2 + 4H^+ + 4e^- = 2H_2O \cdot (E_0 = +1.229 \ V) \tag{2}$$

$$O_2 + 2H^+ + 2e^- = H_2O_2 \ (E_0 = +0.682 \ V) \tag{3}$$

Zhang et al. [15] also observed a rise in the open circuit potential of polyaniline film with time in 0.1 M KNO$_3$ solution. They attributed this rise to ORR. Khomenko et al. [16] studied the autogenous oxidation of many conducting polymers including polyaniline in aqueous HCl and found that ORR is responsible for this phenomenon and that the product of the reaction is H$_2$O$_2$. To test whether hydrogen peroxide is indeed formed during the

OCP experiment on polyaniline in sulfuric acid, we tested the acid solution for detection of $H_2O_2$ at the end of OCP experiment. No hydrogen peroxide was detected in the solution by the starch iodide test [17,18]. We also see from Figure 3 that the equilibrium potential of the film with the lowest mass density is 0.51 $V_{SCE}$, (i.e., +0.752 $V_{RHE}$) which is well above the standard reduction potential for the formation of $H_2O_2$ from oxygen. Hence, we believe that the reduction of oxygen to water (reaction 2) is the only reaction in our experiments. A possible reason for the difference between our results and those of Khomenko et al. [16] is that they have used oxygen-saturated acid electrolyte while we have only carried out surface aeration. We suspect that the $H_2O_2$ mechanism may become important at a high concentration of oxygen in the solution.

In Figure 5, we have plotted OCP curves for a film during two consecutive experiments, the first without nitrogen bubbling (Figure 5a) and the second with oxygen free nitrogen bubbling (Figure 5b). The case without nitrogen bubbling corresponds to surface aeration while that with nitrogen bubbling corresponds to the oxygen free condition.

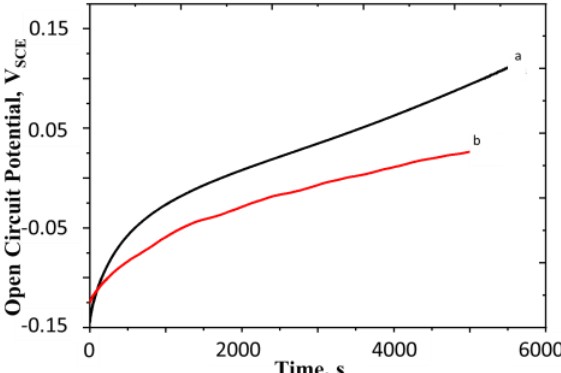

**Figure 5.** Open circuit potential of polyaniline with and without nitrogen bubbling (**a**) OCP curve with surface aeration; (**b**) OCP curve with nitrogen bubbling through the solution. Concentration of $H_2SO_4$ = 0.5 M, surface mass density of the film = 16.3 g m$^{-2}$, temperature = 25 °C. All potentials are reported with respect to SCE.

It is seen from Figure 5b that OCP rises even in the absence of oxygen, albeit the rise is slower and a lower value of the equilibrium potential is attained. This implies that the oxidation of polymer occurs in the absence of oxygen. This observation motivates us to term this oxidation process as autogenous (self-propelled). We discuss the mechanism for this process in detail in Section 4.1.

### 3.3. Determination of the Rate of Oxidation of Polyaniline

In order to determine the rate of the autogenous oxidation of polyaniline film at different electrode potentials, we adopted the following strategy. The electrode potential is electrochemically brought to a value which is greater than the potential at which the oxidation current ($I_O$) is to be estimated. Linear sweep chronopotentiometry is then performed on the film where the initial current is cathodic and its magnitude is reduced with time at a constant current scan rate. The potential of the film is monitored with time. In Figure 6, a typical chronopotentiogram for this experiment is shown. As seen from the figure the potential initially decreases, passes through a minimum, and then increases. We call the point at which the minimum occurs as the turning point. The turning point is marked by a vertical line on the chronopotentiogram.

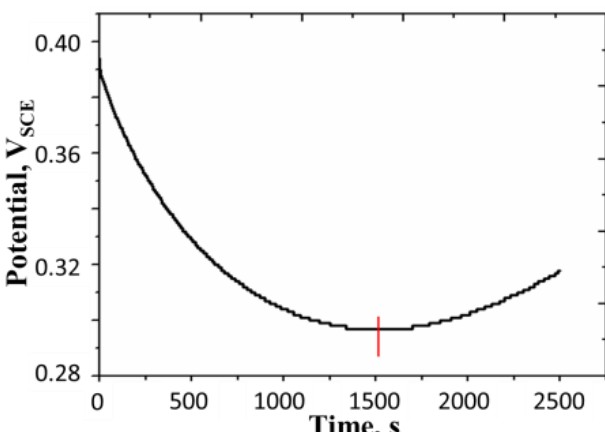

**Figure 6.** Linear current sweep chronopotentiogram performed on a polyaniline film. Concentration of $H_2SO_4$ = 0.5 M, surface mass density of the film = 16.41 g m$^{-2}$, temperature = 25 °C; initial current= −0.25 mA, current scan rate: 10$^{-4}$ mA s$^{-1}$, turning time = 1521 s. All potentials are reported with respect to SCE.

We explain the reason for this turning point as follows with reference to Scheme 1. A thick polyaniline film is deposited on a platinum substrate electrode, which is subjected to linear sweep chronoamperometry. Thus, the cathodic chronoamperometric current (black arrow) reduces in magnitude linearly from 'a' to 'e' as shown in Scheme 1. On the other hand, the autogenous oxidation current (red arrow), which is anodic, is of constant magnitude. The cathodic current tries to reduce the potential of the film. This tendency is opposed by the autogenous oxidation current, which tries to raise the potential of the electrode. The net current, $i$, is, therefore, the difference between the imposed cathodic current $I_C$ and the oxidation current $I_O$. Initially, the imposed cathodic current is greater in magnitude than the autogenous oxidation current (see a and b), that is, $i < 0$. At the turning point, the autogenous oxidation current becomes equal to the cathodic current (see c) so that the net current reduces to zero, or $i = 0$. After the turning point, the autogenous oxidation current surpasses the imposed cathodic current in magnitude (see d and e), so that the net current becomes anodic, that is, $i > 0$. As long as the net current is cathodic, the potential decreases with time. However, when the value of $I_C$ falls below that of $I_O$ the net current becomes anodic. This causes a rise in the electrode potential. The turning point of the potential is that point at which $I_C = I_O$.

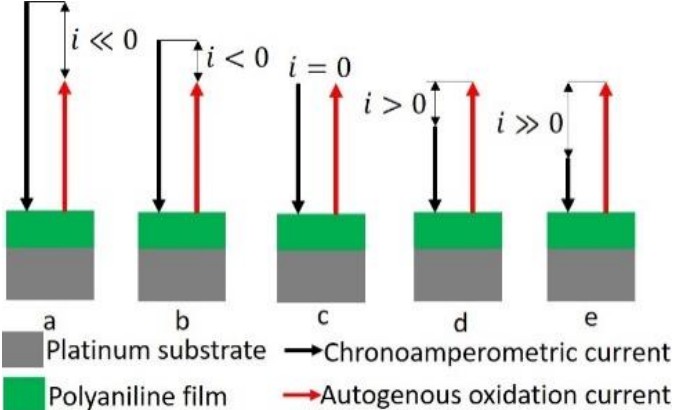

**Scheme 1.** Schematic representation of the turning potential method for determination of autogenous oxidation current.

$I_O$ is estimated from the turning time $t_t$, initial imposed current $I_C^0$ and the current scan rate $v_I$, using the equation

$$I_O = I_C^0 - v_I t_t \tag{4}$$

In Figure 7, we have plotted $i_O$, autogenous oxidation current density (estimated by the procedure described above) against the electrode potential, for different surface mass densities of the film.

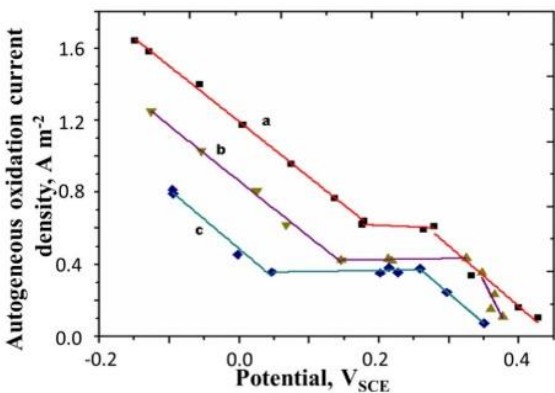

**Figure 7.** Polyaniline oxidation current density vs. potential plot. Concentration of $H_2SO_4$ = 0.5M, Temperature = 25 °C. All potentials are with respect to SCE. Surface mass densities of polyaniline are-curve a: 0. 645 g m$^{-2}$; curve b: 6.92 g m$^{-2}$; curve c: 16.4 g m$^{-2}$.

The plot consists of three regions. There is a plateau region in the middle, flanked by two linear regions. The oxidation current density $i_O$ is seen to decrease with the increase in the electrode potential as well as the surface mass density of the polymer. We will explain these trends later.

### 3.4. Falling Open Circuit Potential Experiments

When the electrode is electrochemically raised above the equilibrium potential and then allowed to relax under the open circuit condition, the potential decreases with time. The open circuit potential curves for polyaniline film, with nitrogen bubbling, and without nitrogen bubbling are shown in Figure 8a,b, respectively.

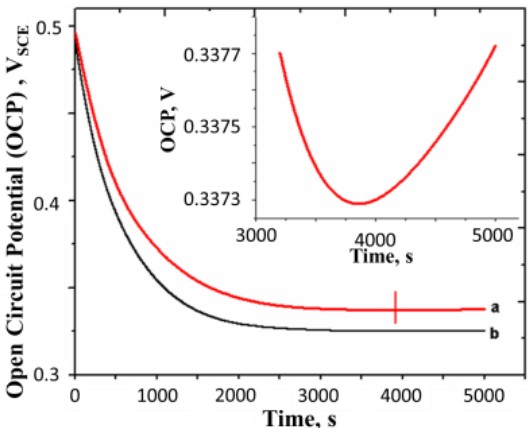

**Figure 8.** Open circuit potential curve of a polyaniline film which is pre-oxidized to a potential much greater than the equilibrium potential (0.39 $V_{SCE}$). (**a**) Without nitrogen bubbling; (**b**) with nitrogen bubbling. Concentration of $H_2SO_4$ = 0.5 M, surface mass density of the film = 16.3g m$^{-2}$ and temperature = 25 °C. All potentials are reported with respect to SCE. Inset shows the OCP in the region around the turning point marked on curve a.

The fall in potential of the film with time indicates the autogeneous reduction of the film. The potential reaches a minimum and then begins to rise. The point at which the turnaround is observed is shown by a vertical line marked on the OCP curve. The turnaround is not visible in the plot, but when the portion around this point is enlarged and replotted in the inset, the turnaround is clearly seen. It is also observed that the fall in potential is faster in the experiment conducted with nitrogen bubbling (i.e., in the absence of oxygen), as seen from Figure 8b.

That the film undergoes the autogeneous reduction is also evident from the anodic chronopotentiogram shown in Figure 9.

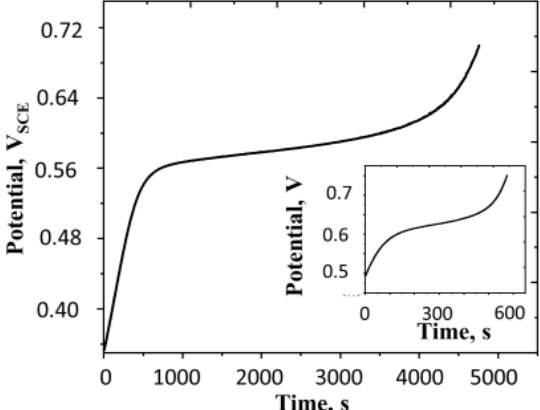

**Figure 9.** Anodic chronopotentiogram of polyaniline film at constant current density of 2.50 A m$^{-2}$. Surface aerated system, concentration of $H_2SO_4$ = 0.5 M, surface mass density of the film = 16.3 g m$^{-2}$ and temperature = 25 °C. All potentials are reported with respect to SCE. The inset shows the chronopotentiogram for the same film at the current density of 7.5 A m$^{-2}$ under otherwise identical conditions.

We see that the autogenous reduction current (which is cathodic) opposes the imposed anodic current and as a result, the rise in the potential of the film is impeded in the mid-region. The reason for the initial and final sharp rise in the potential is explained later.

We have applied the turning point technique (discussed with reference to Figure 5). In this case, the anodic linear sweep chronopotentiometry was used. A typical turning point curve is shown in Figure 10.

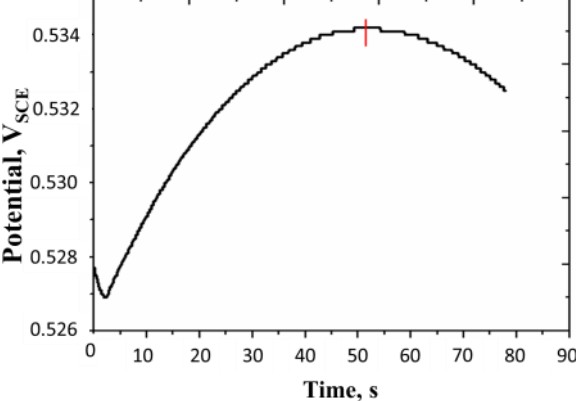

**Figure 10.** Linear current sweep chronopotentiogram of a polyaniline film. Surface aerated system, concentration of $H_2SO_4$ = 0.5 M, surface mass density of the film = 16.41 g m$^{-2}$, temperature = 25 °C. Initial current = +0.8 mA, current scan rate: $5 \times 10^{-3}$ mA s$^{-1}$, turning time = 52 s. All potentials are reported with respect to SCE.

Using this technique, we measure the reduction current density as a function of the potential of the electrode. A complete plot, marking the zones of oxidation and reduction along with the intermediate region, where both reactions occur, is shown in Figure 11.

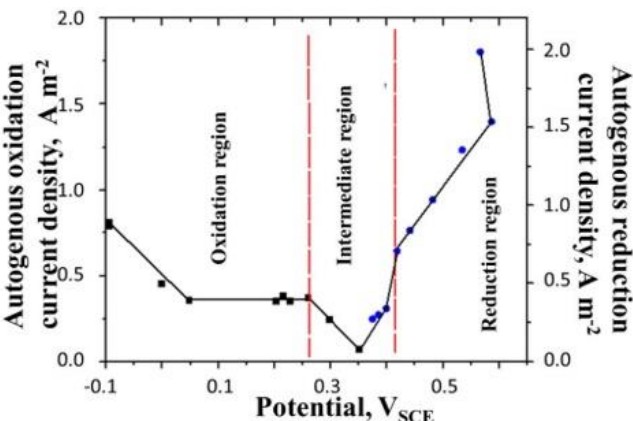

**Figure 11.** Variation of the oxidation and reduction current density in polyaniline film. Surface aerated system, concentration of $H_2SO_4$ = 0.5 M, surface mass density of the film = 16.4 g m$^{-2}$ and temperature = 25 °C. All potentials are reported with respect to SCE.

It is seen from this figure that the reduction current density $i_R$ increases with the increase in the electrode potential. In the vicinity of the equilibrium potential, the reduction current density is nearly zero. Moreover, the maximum reduction current density is almost twice as large as the maxiumum oxidation current density, indicating that the autogeneous reduction reaction is in general faster than the autogeneous oxidation reaction.

## 4. Discussion

### 4.1. Mechanism of Oxidation of Polyaniline

We propose the following mechanism for autogeneous oxidation of polyaniline in the absence of oxygen.

$$-NH- +H^+ \rightarrow -N^+H_2- \tag{5}$$

$$-N^+H_2- \leftrightarrow -N^{\cdot+}H- + H^\cdot \tag{6}$$

The first step involves the protonation of the amine nitrogen of polyaniline. This step is followed by the generation of a polaron and a hydrogen radical, H$^\cdot$. Oxygen reacts with H$^\cdot$ to form water.

$$4H^\cdot + O_2 \rightarrow 2H_2O \tag{7}$$

Hence oxygen merely hastens the autogeneous oxidation process.

There is also a possibility of a parallel mode where oxygen can directly oxidize the reduced form of polyaniline to produce polarons as suggested by Khomenko et al. [4].

$$4-NH_2^+ - +O_2 \rightarrow 4-N^{\cdot+}H- +2H_2O + 4e^- \tag{8}$$

Electrons released through this reaction may react with hydrogen ions to produce H$^\cdot$ radicals.

In the absence of oxygen, we expect hydrogen radicals to accumulate on the aromatic rings of polyanline, thus reducing the mesomeric stabilization of polarons. Parallely, they can combine with each other and produce hydrogen gas. The overall effect of these processes is not only to reduce the equilibrium potential but also to slow down the approach to it.

Using this mechanism, we can explain the trends observed in Figure 7. We assume that the oxidation current (which is proportional to the rate of generation of polarons) is

controlled by reaction 5 and is the first order in the surface density of protonated amines in the film, denoted by $[A^+]$.

$$i_O = kF[A^+] \tag{9}$$

Here, $i_O$ is the current density for polyaniline oxidation and $k$ is the first order rate constant for the reaction.

Concentration of polarons in the film increases linearly with electrode potential (since we have shown that the differential capacitance of the film is constant in Figure 1). The increase in the potential of the film causes an increase in the linear density of the positive charges on the polymer chains. This causes the exclusion of hydrogen ions from the double layer surrounding the chains. As a result, there will be a reduction in the extent of protonation of the amine nitrogen. That the degree of protonation of polyaniline decreases with the increase in the potential is shown by Orata and Buttry using peizoelectric quartz crystal microbalance [19]. We assume that the extent of protonation reduces linearly with an increase in the linear charge density of the chains in the low potential region and hence we express the relation between the concentration of the protonated amine groups, $[A^+]$, and that of polarons $[P]$ as follows

$$[A^+] = [A^+]_0 - \alpha[P] \tag{10}$$

where all concentrations are expressed per unit area of the electrode. $[A^+]_0$ is the maxiumum attainable extent of protonation and the term $\alpha$ is a proportionality constant which reflects the increase in potential of the polymer due to polaron formation.

We note that,

$$\frac{d\sigma}{dE} = c_f \tag{11}$$

where $\sigma$ is the electric charge in the film per unit area of the electrode and $E$ is the electrode potential. We can relate $\sigma$ to the concentration of polarons as follows:

$$\sigma = F[P] \tag{12}$$

where $F$ is the Faraday constant.

Combining Equations (9)–(12), we obtain

$$-F\frac{d[A^+]}{dE} = \alpha c_f \tag{13}$$

Combining Equations (9) and (13), we obtain

$$-\frac{di_O}{dE} = k\alpha c_f \tag{14}$$

Since $k$, $\alpha$ and $c_f$ are constant, the slope of the plot of $i_O$ versus $E$ is linear and constant at lower potentials.

At higher potentials, the surface charge density on the chain becomes so high that bisulfate ions ($HSO_4^-$) begin to condense on the chains [20]. This does not allow the linear charge density to increase beyond a certain limiting value. Beyond this limiting potential, the extent of protonation does not change and hence current density attains a plateau.This continues until a potential is reached where the bipolarons begin to dominate over polarons as charge carriers. Since bipolarons have low capacity to store charge, the current density falls with a further increase in the potential [21].

The slope of the $i_O$-$E$ plot (Figure 7) in the low potential region is seen to be independent of the mass density of polymer. This slope is equal to $-k\alpha c_f$ from Equation (14). The specific capacitance of the film, $c_f$, is expected to increase proportionately with the mass density. Hence, to achieve a constant slope, the product $k\alpha$ must decrease. With the increase in chain length, the stability of protonated amine nitrogens in polyaniline is expected to increase due to the mesomeric effect (exchange of protons among protonated

and unprotonated nitrogens). This is also consistent with the increase in conductivity of the polymer with the increase in the acidity of the medium [22]. This increases the activation energy for conversion of protonated amines to polarons, thereby decreasing the rate constant, $k$. As the chain length increases, $\alpha$ also decreases because the resonance stabilization opposes the increase in potential due to autogenous oxidation. The effect of the decrease in both $k$ and $\alpha$ seems to counteract the effect of increase in $c_f$ which explains the invariance in the slope of the $i_O$-$E$ line.

### 4.2. Mechanism of Reduction

We propose the following mechanism for the reduction reaction of the polyaniline film. It consists of the following step:

$$-N^{\cdot+}H - +H_2O \rightarrow HO^{\cdot} + -N^{+}H_2- \qquad (15)$$

Reaction 15 produces a hydroxyl radical $HO^{\cdot}$, thereby reducing the polaron to protonated amine. The reduction of polarons in the film causes a reduction in its potential as the protonated amine groups do not contribute to the pseudocapacitance of the film.

Another important point to note is that $HO^{\cdot}$ radical generated by reaction 15 forms a stable charge transfer complex with polarons present in the film, thereby stabilizing it. These $HO^{\cdot}$ radicals accumulate in the film with time until the OCP enters the intermediate region as in Figure 11. In this region, both the oxidation and reduction reactions occur simultaneously. $H^{\cdot}$ radicals generated by the oxidation reaction destroy $HO^{\cdot}$ radicals present in the film according to the reaction.

$$H^{\cdot} + HO^{\cdot} \rightarrow H_2O \qquad (16)$$

We can explain the trend in Figure 8 with the help of this mechanism. At the beginning of the falling OCP experiment, the film has a high concentration of polarons (which were generated during the electrochemical pre-oxidation of the film). As the reaction 15 occurs, polarons are reduced and potential falls. Moreover, $HO^{\cdot}$ form stable charge tranfer complexes with the polarons. The lower chemical potential of the stabilized polaron results in an increase in the film capacitance. This higher capacitance allows the film to carry more polarons at a given potential (where some of them are occupied by $HO^{\cdot}$ radicals) compared to the case where the film has no $HO^{\cdot}$ radicals. This is the reason why OCP falls below 0.39 $V_{SCE}$. When the potential is in the intermediate region as in Figure 11, $H^{\cdot}$ radicals are generated, which destroy $HO^{\cdot}$ radicals. In the presence of oxygen, the rate of generation of hydrogen radicals is enhanced as shown earlier. Hence with the passage of time, the excess $HO^{\cdot}$ radicals are destroyed by $H^{\cdot}$ radicals, thereby reducing the film capacitance. This would make the potential to turn around after some time as shown in the inset of Figure 8. Eventually, the film attains the same equilibrium potential as observed during the rising potential experiment. In the absence of oxygen, $HO^{\cdot}$ radicals produced are not destroyed by the reaction with $H^{\cdot}$ radicals. Hence, the film capacitance is even higher than in the surface aerated mode. Thus, the plot in Figure 8b falls below Figure 8a.

We can also explain the trend observed in Figure 9. Here, we see the slow varying potential region in the middle, flanked by two fast varying potential regions. The initial fast variation of potential is caused by the fact that at lower potentials, the rate of autogeneous reduction is slow due to the low concentration of polarons in the film and hence the imposed anodic current dominates the overall current. As a result, the potential rise is fast. In the middle region, both the imposed current, and the reduction current approach each other; hence, the net current is very small, but anodic, which causes a slow rise in the potential. However, during this period, there is the accumulation of hydroxyl radicals in the film, which block the polarons and make them unavailable for reduction. Only the free polarons are available for the reduction. These decrease with time as an increasing fraction of the polarons is blocked by $HO^{\cdot}$. Hence beyond a point, the rate of generation of amines

by the reduction of polarons becomes lower than the rate at which they are oxidized by the imposed current, and this situation results in a steep rise in the potential.

It is illustrative to approximately estimate the extent of oxidation of the film at which steep potential rise is exhibited in Figure 9. From the initial slope of the chronopotentiogram, the capacitance of the film can be estimated as 7000 F m$^{-2}$. Assuming that the film has no polarons at the potential of $-0.2$ V$_{SCE}$, noting that at the beginning of the experiment is $+0.35$V$_{SCE}$, the charge initially present in the film is 3.85 kC m$^{-2}$. During the period from the beginning of the experiment until the beginning of the steep potential rise (4000 s), the charge accummulated in the film at the charging current of 2.50 A m$^{-2}$ is 10 kC m$^{-2}$. Hence, the total charge accummulated in the film at that point is 13.85 kC m$^{-2}$. This corresponds to the oxidation of 0.144 mol m$^{-2}$ of amine out of the total 0.179 mol m$^{-2}$ of amine present in the film, or 80.5% oxidation. Thus, we see that a steep rise in potential occurs when the major fraction of the film is oxidized.

When the imposed anodic current is increased to 7.50 A m$^{-2}$ (see the inset of Figure 9), the steep potential rise occurs at a lower conversion since the higher concentration of free polarons are needed for the reduction reaction in order to match the imposed current. The calculations show that the steep rise occurs when the charge in the film is 7.58 kC m$^{-2}$. This corresponds to the extent of oxidation of 43.9% of the film.

We can also explain the linear trend of the reduction current in Figure 11 as the potential of the film is increased. If we assume the rate of the reduction reaction to be the first order in the concentration of polarons, then, since, the capacitance of the film is constant, the rate of reduction is expected to vary linearly with the potential in the major portion of the reduction curve. This is consistent with the trend observed in Figure 11.

## 5. Conclusions

In this work, we have observed that polyaniline film undergoes both autogenous oxidation and reduction reactions. Using the turning potential analysis, we have been able to determine the kinetics of these reactions. We have made two important observations. The first is that the autogenous oxidation of polyaniline can occur even in the absence of oxygen. However, the presence of oxygen accelerates the autogenous oxidation. Second, the autogenous reduction reaction involves the oxidation of water molecules with the concomitant reduction of polarons. This process generates hydroxyl radicals which form stable complexes with aromatic rings of the polymer. This is the reason why the rate of autogenous reduction is significantly higher than the rate of autogenous oxidation.

**Author Contributions:** Conceptualization, V.J. and A.C.; methodology, A.S. (Amrita Singh); software, A.S. (Amrita Singh); validation, A.S. (Amrita Singh) and A.S. (Arindam Sarkar); formal analysis, V.J.; investigation, A.S. (Amrita Singh); resources, V.J.; data curation, V.J. and A.S. (Arindam Sarkar); writing—original draft preparation, A.C.; writing—review and editing, A.C.; visualization, A.C.; supervision, V.J. and R.K.; project administration, A.C.; funding acquisition, A.C. All authors have read and agreed to the published version of the manuscript.

**Funding:** This research was funded by the Science and Engineering Research Board (SERB), Government of India, Project no. YSS/2015/001212.

**Institutional Review Board Statement:** Not applicable.

**Informed Consent Statement:** Not applicable.

**Data Availability Statement:** Not applicable.

**Conflicts of Interest:** The authors declare no conflict of interest.

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
