# Peer review of "Autogenous Oxidation/Reduction of Polyaniline in Aqueous Sulfuric Acid"

_processes, doi:10.3390/pr10030443_

Round 1

Reviewer 1 Report

This work studies both autogenous oxidation and reduction reactions of polyaniline film in aqueous sulfuric acid. The kinetics of these reactions are determined by using the turning potential analysis. This work is recommended for minor revision and the following questions should be addressed.

A fundamental discussion for this work can be schematically illustrated and the relative scheme is encouraged.

Many misuse of the symbols are found in the manuscript, such as A.m-2, F.g-1, etc.

The references should be cited more properly. Only 2 recently published works are cited and most of them are before 1995.

The formation of the references need improvement.

Author Response

Authors' response to comments by reviewer 1:

  1. A scheme illustrating the turning potential analysis has been added to the manuscript. It is discussed in detail.
  2. The symbols for the units have been corrected to A.m-2, F.g-1, C.m-2, etc.
  3. We have extended the introduction to emphasize the originality of our work and its usefulness. To this end we have included 12 additional references in the manuscript.
  4. We have reformatted the references as per the guidelines.

Reviewer 2 Report

The manuscript describes the autogenous oxidation and reduction of polyaniline film with the turning potential analysis. This work is quite interesting and the draft was well prepared. However, I think there are several points to be revised.

In figure 1, It is not unclear that the relationship between the changed shape of cyclic voltammograms and merging of the energy bands. The y-axis ranges of all figures must be matched to the same value. In this manuscript, as the mass density increases, the peak increases. However, the data is unreliable without the scan rate. So, the value of scan rate is needed. Additionally, the detail contents such as y-axis font size or graph size need to be unified.

The figure 3 shows that the equilibrium potential related on surface mass density of polyaniline. The values of mass density should be matched the values of figure 1. Overall, the mass density is different for each figure. The additional explanation of the 16 g.m-2 from figure 5 is needed.

In my opinion this manuscript is lacking sufficient justification of the pertinence of the attempted approach. The experiment design seemed to be conceived in a random way with many different conditions. The references are insufficient to support the contents of this manuscript.

Author Response

Authors' response to comments by Reviewer 2:

  1. Figure 1 has been replotted in two parts: a and b. Figure 1a displays cyclic voltammograms of five polyaniline films in the order of increasing thickness. The merging of the peaks to form a leaf-shaped voltammogram is clearly seen. Figure 1b displays the corresponding cyclic coulograms. This method of plotting unifies the y-axis for all cyclic voltammograms and cyclic coulograms, so that they can be easily compared. The scan rate (50mV/s) is also included in the figure caption.
  2. Figure 3 shows equilibrium potentials of films displayed in Figure 1, that is, they have the corresponding mass densities. Five films out of six films shown in Figure 3 have been displayed in the CVs of Figure 1. The film which has mass density 13g/m2 has been excluded because the CV is very similar to the 16.4 g/m2 film.
  3. We have discussed the result of Figure 5 in section 4.1. We have referred to it in the manuscript.
  4. We have extended the introduction to this manuscript to emphasize the usefulness and originality of our work. We have also included 12 recent references. 

Round 2

Reviewer 2 Report

This manuscript was revised well and the work is suitable for publication in this journal.